# Anti-Inflammatory Properties of Novel 1,2-Benzothiazine Derivatives and Their Interaction with Phospholipid Model Membranes

**DOI:** 10.3390/membranes14120274

**Published:** 2024-12-18

**Authors:** Berenika M. Szczęśniak-Sięga, Jadwiga Maniewska, Benita Wiatrak, Tomasz Janek, Paulina Nowotarska, Żaneta Czyżnikowska

**Affiliations:** 1Department of Medicinal Chemistry, Faculty of Pharmacy, Wroclaw Medical University, Borowska 211, 50-556 Wroclaw, Poland; jadwiga.maniewska@umw.edu.pl; 2Department of Pharmacology, Faculty of Medicine, Wroclaw Medical University, J. Mikulicza-Radeckiego 2, 50-345 Wroclaw, Poland; benita.wiatrak@umw.edu.pl; 3Department of Biotechnology and Food Microbiology, Faculty of Biotechnology and Food Science, Wroclaw University of Environmental and Life Sciences, Chełmońskiego 37, 51-630 Wroclaw, Poland; tomasz.janek@upwr.edu.pl; 4Department of Biostructure and Animal Physiology, Wroclaw University of Environmental and Life Sciences, Norwida 25/27, 50-375 Wroclaw, Poland; paulina.nowotarska@upwr.edu.pl; 5Department of Basic Chemical Sciences, Faculty of Pharmacy, Wroclaw Medical University, Borowska 211a, 50-556 Wroclaw, Poland

**Keywords:** 1,2-benzothiazine derivatives, drug–membrane interaction, anti-inflammatory activity, antioxidant properties

## Abstract

The design of novel anti-inflammatory drugs remains a critical area of research in the development of effective treatments for inflammatory diseases. In this study, a series of 1,2-benzothiazine was evaluated through a multifaceted approach. In particular, we investigated the potential interactions of the potential drugs with lipid bilayers, an important consideration for membrane permeability and overall pharmacokinetics. In addition, we evaluated their ability to inhibit cyclooxygenase 1 and cyclooxygenase 2 activity and selectivity using both a cyclooxygenase inhibition assay and molecular docking simulations. To evaluate their therapeutic potential, we performed in vitro assays to measure cytokine mRNA expression in inflamed cells. The antioxidant activity was evaluated using both in vitro assays, such as 2,2-diphenyl-1-picrylhydrazyl and 2,2-azino-*bis*-3-ethylbenzothiazoline-6-sulphonic acid scavenging, to determine the compounds’ capacity to neutralize free radicals and reduce oxidative stress. Theoretical calculations, including density functional theory, were used to predict the reactivity profiles of the compounds.

## 1. Introduction

Cyclooxygenase (E.C. 1.14.99.1., prostaglandin cyclic peroxide synthase, PGHS, COX) is an enzyme belonging to the metalloproteinase family, located in the lumen of the endoplasmic reticulum, the Golgi apparatus, and the nuclear membrane of the cell, embedded with hydrophobic fragments in the lipid bilayer of the membrane. COX catalyzes the process of prostanoid synthesis, i.e., prostaglandins, prostacyclin, and thromboxane from arachidonic acid. Three isoforms of this enzyme have been identified so far: COX-1, COX-2, and COX-3 [1]. Since COX-2 is the isoform induced by the inflammation process, great efforts to find selective COX-2 inhibitors have been made between the last decade of 1900 and the beginning of 2000, resulting in several compounds being approved worldwide (e.g., celecoxib). Some of these were subsequently withdrawn from the market due to cardiovascular toxicity (e.g., rofecoxib and valdecoxib) [2]. However, since inflammation has been linked to cancer development, the search for new, safer anti-inflammatory drugs has become even more important [3,4,5,6]. It has been established that in the process of carcinogenesis, overexpression, increased secretion, or inappropriate activation of proinflammatory mediators, such as COX-2, prostaglandins, cytokines, inducible nitric oxide synthase and nitric oxide, and many other intracellular signaling molecules, facilitate tumor promotion and progression [7,8]. In addition, inflammation induces the formation of reactive oxygen species (ROS) and nitrogen species (RNS), which cause damage to very important cellular elements such as DNA, proteins, or lipids, which may directly or indirectly contribute to neoplastic transformation [9,10]. Taking these data into account, finding new COX-2 inhibitors that also have free radical scavenging properties would be beneficial both in inflammatory diseases and in cancer treatment.

In our previous studies, we obtained a series of 1,2-benzothiazine derivatives with anti-inflammatory and analgesic activity confirmed in in vivo tests on mice [11]. In addition, these compounds were devoid of ulcerogenic activity characteristic of most non-steroidal anti-inflammatory drugs (NSAIDs). These compounds were modified in relation to the model drug—meloxicam (Figure 1), in positions 2 and 3 of the thiazine ring. The modification in position 2 seems to be more important for the activity of the new compounds because it consisted of replacing the small methyl substituent in meloxicam with a large, extended arylpiperazine group. According to Hatnapure, the high electron-donating capacity of the piperazine moiety can be clearly correlated with the high anti-inflammatory activity of the compounds, which was consistent with many other scientific studies [12,13,14,15,16,17].

One of these tested and non-toxic compounds was a 1,2-benzothiazine derivative containing a heterocyclic pyrimidine ring and a bromine atom in its structure (Figure 2a). We decided to check whether these substituents significantly influenced the anti-inflammatory activity of this compound, and therefore, we planned to obtain a new derivative in which the pyrimidine ring was replaced with a benzene ring (Figure 2b). Another planned modification was to replace the acetyl linker between the thiazine and piperazine nitrogen atoms with a propyl linker because compounds with such a linker also showed strong anti-inflammatory activity and did not cause ulcer formation, as shown by our previous studies (Figure 2c) [11]. Subsequent modifications included replacing the bromine atom with other electron-withdrawing substituents, such as a fluorine atom, or electron-donating substituents such as a methyl or methoxy group. On the other hand, the pyrimidine ring was replaced with a pyridine ring to assess the effect of the type of ring on the anti-inflammatory activity of the compound (Figure 2d).

The primary objective of this study was to determine whether compounds based on an arylpiperazine-1,2-benzothiazine core could serve as a lead structure for new, safer anti-inflammatory and at the same time antioxidant compounds. Hence, we performed an in vitro evaluation of the impact of the tested compounds on the activity of COX-2 and COX-1 to assess the selectivity ratio. The anti-inflammatory effect of the new compounds was also confirmed in the MTT and DCF-DA assays on normal human dermal fibroblast (NHDF) cells exposed to lipopolysaccharide (LPS). The radical scavenging activity was evaluated with two different in vitro measurements, using 2-diphenyl-1-picrylhydrazyl (DPPH) and 2,2′-azino-bis(3-ethylbenzothiazoline-6-sulphonic acid (ABTS). An additional goal was to assess the ability of designed compounds to interact with phospholipid bilayers because cyclooxygenase is associated with biological membranes. Experimental results were supported by theoretical calculations. The global reactivity parameters were obtained based on the density level of theory. The binding mode of designed compounds to both cyclooxygenase isoforms was established by using molecular docking.

## 2. Materials and Methods

### 2.1. Chemistry

All commercial chemicals were used as supplied unless otherwise indicated. ^1^H and ^13^C NMR spectra were recorded on a Bruker 300 MHz spectrometer using CDCl_3_ as a solvent. Chemical shifts for proton nuclear magnetic resonance (^1^H NMR) spectra are reported in parts per million (ppm) relative to the signal of tetramethylsilane at 0 ppm (internal standard). Splitting patterns are designated as follows: s, singlet; brs, broad singlet; d, doublet; t, triplet; q, quartet; m, multiplet. Chemical shifts for carbon nuclear magnetic resonance (^13^C NMR) spectra are reported in parts per million (ppm) relative to the center line of the CDCl_3_ triplet at 77.02 ppm. FT-IR spectra were recorded on a Perkin-Elmer Spectrum Two UATR FT-IR spectrometer. Mass data were acquired on a Bruker Daltonics micrOTOF-Q Mass Spectrometer in a positive ion mode with flow injection electron spray ionization (ESI). The elemental analyses were carried out on a Carlo Erba NA 1500 analyzer and were within ±0.4% of the theoretical value. Melting points were determined in open glass capillaries using a MEL-TEMP melting-point apparatus and were uncorrected. Reaction progress was monitored using thin-layer chromatography (TLC) performed on silica gel 60 F254 pre-coated aluminum sheets. Spots were visualized under a 254 nm UV lamp. A rotary evaporator under reduced pressure conditions was used to concentrate the reaction solutions. Solvents were of reagent grade and purified with standard methods when necessary.

Synthesis and experimental data of compounds **3a**–**d** and **4a**–**d** were previously reported [18,19], and compounds **5**, **6,** and **7** were reported also [20,21].

Synthesis and Experimental Data of Compounds **BS23**–**BS30**

A solution of 5 mL of sodium ethoxide (2.3%) was added to the stirred mixture of 5 mmol of corresponding 1,2-benzothiazines **4a**–**d** in 20 mL of anhydrous ethanol. Then, 5 mmol of compound **5**, **6,** or **7** was added and refluxed with stirring for 10 h. When the reaction ended, which was controlled on TLC plates, ethanol was distilled off, and the residue was treated with 50 mL of chloroform and insoluble materials (e.g., sodium chloride) were filtered off. The filtrate was then evaporated, and the residue was purified by crystallization from ethanol to give desirable products **BS23**–**BS30** with a yield of 25–50%.

3-(4-Methoxybenzoyl)-4-hydroxy-2-{3-[4-(2-pyridyl)-1-piperazinyl)propyl]}-2H-1,2-benzothiazine 1,1-dioxide **BS23**

Yellow powder, 33.05% yield, mp 107–111 °C; FT-IR (cm^−1^): 1605 (C=O), 1345, 1170 (SO_2_). ^1^H NMR (300 MHz, CDCl_3_) *δ* (ppm): 1.41 (brs, 2H, CH_2_CH_2_CH_2_), 2.11–2.30 (m, 6H, CH_2_N(CH_2_)_2_), 3.25–3.48 (m, 6H, CH_2_N(CH_2_)_2_), 3.88 (s, 3H, OCH_3_), 6.57–8.21 (m, 12H, ArH), 15.75 (s, 1H, OH*_enolic_*). ^13^C NMR (300 MHz, CDCl_3_) *δ* (ppm): 190.01, 163.46, 158.98, 147.86, 138.08, 137.64, 133.46, 132.83, 131.98, 128.97, 127.98, 127.83, 123.38, 116.72, 113.92, 107.12, 55.53, 55.06, 52.26, 51.71, 44.41, 23.11. HRMS (ESI) calculated for C_28_H_30_N_4_O_5_S [M+H]^+^ 535.2010; found the following: 535.1957. Analysis calculated for C_28_H_30_N_4_O_5_S (534.63); C, 62.90; H, 5.66; N, 10.48; found the following: C, 62.88; H, 5.80; N, 10.42.

3-(4-Methoxybenzoyl)-4-hydroxy-2-{3-[4-(2-pyrimidyl)-1-piperazinyl)propyl]}-2H-1,2-benzothiazine 1,1-dioxide **BS24**

Yellow powder, 36.10% yield, mp 60–65 °C; FT-IR (cm^−1^): 1605 (C=O), 1345, 1175 (SO_2_). ^1^H NMR (300 MHz, CDCl_3_) *δ* (ppm): 1.44 (brs, 2H, CH_2_CH_2_CH_2_), 2.12–2.29 (m, 6H, CH_2_N(CH_2_)_2_), 3.23 (brs, 2H, -CH_2_-), 3.70–3.83 (brs, 4H, N(CH_2_)_2_), 3.87 (s, 3H, OCH_3_), 6.49–8.29 (m, 11H, ArH). ^13^C NMR (300 MHz, CDCl_3_) *δ* (ppm): 189.86, 169.20, 163.45, 161.33, 157.74, 137.96, 133.47, 132.87, 131.99, 128.97, 127.94, 127.87, 123.47, 116.85, 113.91, 110.32, 55.53, 55.10, 52.31, 51.83, 42.61, 23.08. HRMS (ESI) calculated for C_27_H_29_N_5_O_5_S [M+H]^+^ 536.1962; found the following: 536.1906. Analysis calculated for C_27_H_29_N_5_O_5_S (535,61); C, 60.55; H, 5.46; N, 13.08; found the following: C, 60.80; H, 5.70; N, 13.35.

3-(4-Methylbenzoyl)-4-hydroxy-2-{3-[4-(2-pyridyl)-1-piperazinyl)propyl]}-2H-1,2-benzothiazine 1,1-dioxide **BS25**

Yellow crystals, 34.14% yield, mp 139–141 °C; FT-IR (cm^−1^): 1600 (C=O), 1350, 1180 (SO_2_). ^1^H NMR (300 MHz, CDCl_3_) *δ* (ppm): 1.42 (brs, 2H, CH_2_CH_2_CH_2_), 2.10–2.32 (m, 6H, CH_2_N(CH_2_)_2_), 2.42 (s, 3H, CH_3_), 3.20 (brs, 2H, -CH_2_-), 3.52 (brs, 4H, N(CH_2_)_2_), 6.58–8.21 (m, 12H, ArH). ^13^C NMR (300 MHz, CDCl_3_) *δ* (ppm): 190.50, 169.98, 158.85, 147.91, 143.71, 138.07, 137.68, 133.58, 132.88, 129.64, 129.25, 128.95, 127.92, 123.52, 117.10, 113.78, 107.16, 55.06, 52.14, 44.23, 23.14, 21.74. HRMS (ESI) calculated for C_28_H_30_N_4_O_4_S [M+H]^+^ 519.2061; found the following: 519.2008. Analysis calculated for C_28_H_30_N_4_O_4_S (518,63); C, 64.84; H, 5.83; N, 10.80; found the following: C, 65.15; H, 5.92; N, 10.66.

3-(4-Fluorobenzoyl)-4-hydroxy-2-{3-[4-(2-pyridyl)-1-piperazinyl)propyl]}-2H-1,2-benzothiazine 1,1-dioxide **BS26**

Beige powder, 50.15% yield, mp 155–158 °C; FT-IR (cm^−1^): 1600 (C=O), 1345, 1175 (SO_2_). ^1^H NMR (300 MHz, CDCl_3_) *δ* (ppm): 1.44 (brs, 2H, CH_2_CH_2_CH_2_), 2.14–2.39 (m, 6H, CH_2_N(CH_2_)_2_), 3.21 (brs, 2H, -CH_2_-), 3.54 (brs, 4H, N(CH_2_)_2_), 6.59–8.21 (m, 12H, ArH). ^13^C NMR (300 MHz, CDCl_3_) *δ* (ppm): 190.00, 170.00, 158.77, 147.85, 138.03, 137.74, 133.72, 132.98, 132.24, 132.12, 128.01, 123.58, 116.94, 115.91, 115.62, 113.84, 107.24, 55.04, 52.20, 44.25, 23.13. HRMS (ESI) calculated for C_27_H_27_FN_4_O_4_S [M+H]^+^ 523.1810; found the following: 523.1759. Analysis calculated for C_27_H_27_FN_4_O_4_S (522.59); C, 62.05; H, 5.21; N, 10.72; found the following: C, 62.19; H, 5.27; N, 10.51.

3-(4-Fluorobenzoyl)-4-hydroxy-2-{3-[4-(2-pyrimidyl)-1-piperazinyl)propyl]}-2H-1,2-benzothiazine 1,1-dioxide **BS27**

Beige powder, 40.40% yield, mp 151–153 °C; FT-IR (cm^−1^): 1600 (C=O), 1340, 1170 (SO_2_). ^1^H NMR (300 MHz, CDCl_3_) *δ* (ppm): 1.42 (brs, 2H, CH_2_CH_2_CH_2_), 2.12–2.31 (m, 6H, CH_2_N(CH_2_)_2_), 3.21 (brs, 2H, -CH_2_-), 3.69–3.79 (m, 4H, N(CH_2_)_2_), 6.49–8.29 (m, 11H, ArH). ^13^C NMR (300 MHz, CDCl_3_) *δ* (ppm): 189.79, 169.78, 163.66, 161.35, 157.75, 138.04, 133.72, 132.97, 132.25, 132.13, 127.99, 123.58, 116.92, 115.92, 115.63, 110.36, 55.04, 52.37, 51.96, 42.66, 23.16. HRMS (ESI) calculated for C_26_H_26_FN_5_O_4_S [M+H]^+^ 524.1762; found the following: 524.1705. Analysis calculated for C_26_H_26_FN_5_O_4_S (523,58); C, 59.64; H, 5.01; N, 13.38; found the following: C, 59,60; H, 5.30; N, 13.31.

3-(4-Bromobenzoyl)-2-[2-(4-phenylpiperazin-1-yl)-2-oxoethyl]-4-hydroxy-2H-1,2-benzothiazine 1,1-dioxide **BS28**

Yellow powder, 36.22% yield, mp 134–138 °C; FT-IR (cm^−1^): 1600, 1660 (C=O), 1340, 1180 (SO_2_). ^1^H NMR (300 MHz, CDCl_3_) *δ* (ppm): 2.98–3.07 (m, 4H, N(CH_2_)_2_), 3.41 (brs, 4H, N(CH_2_)_2_), 4.37–4.39 (m, 2H, CH_2_CO), 7.00–8.26 (m, 13H, ArH), 15.46 (s, 1H, OH*_enolic_*).^13^C NMR (300 MHz, CDCl_3_) *δ* (ppm): 187.09, 171.86, 164.35, 138.63, 134.04, 133.39, 132.65, 132.04, 130.64, 129.53, 129.39, 127.77, 122.36, 117.50, 116.15, 58.42, 51.94, 50.32, 44.52, 41.06, 18.43. HRMS (ESI) calculated for C_27_H_24_BrN_3_O_5_S [M+H]^+^ 584.0676; found the following: 584.0616. Analysis calculated for C_27_H_24_BrN_3_O_5_S (582,46); C, 55.68; H, 4.15; N, 7.21; found the following: C, 55.57; H, 4.47; N, 6.88.

3-(4-Bromobenzoyl)-4-hydroxy-2-{3-[4-(2-pyridyl)-1-piperazinyl)propyl]}-2H-1,2-benzothiazine 1,1-dioxide **BS29**

Yellow powder, 25.20% yield, mp 87–90 °C; FT-IR (cm^−1^): 1602 (C=O), 1350, 1175 (SO_2_). ^1^H NMR (300 MHz, CDCl_3_) *δ* (ppm): 1.48 (brs, 2H, CH_2_CH_2_CH_2_), 2.21–2.45 (m, 6H, CH_2_N(CH_2_)_2_), 3.20 (brs, 2H, -CH_2_-), 3.60 (brs, 4H, N(CH_2_)_2_), 6.60–8.20 (m, 12H, ArH). ^13^C NMR (300 MHz, CDCl_3_) *δ* (ppm): 158.66, 147.86, 137.78, 133.77, 133.05, 131.82, 131.01, 128.09, 123.73, 117.31, 113.96, 107.30, 55.13, 52.19, 44.11, 23.18. HRMS (ESI) calculated for C_27_H_27_BrN_4_O_4_S [M+H]^+^ 585.0992; found the following: 585.0924. Analysis calculated for C_27_H_27_BrN_4_O_4_S (583.50); C, 55.58; H, 4.66; N, 9.60; found the following: C, 55.62; H, 4.74; N, 9.33.

3-(4-Bromobenzoyl)-4-hydroxy-2-{3-[4-(2-pyrimidyl)-1-piperazinyl)propyl]}-2H-1,2-benzothiazine 1,1-dioxide **BS30**

Yellow powder, 37.30% yield, mp 152–156 °C; FT-IR (cm^−1^): 1600 (C=O), 1340, 1170 (SO_2_). ^1^H NMR (300 MHz, CDCl_3_) *δ* (ppm): 1.52 (brs, 2H, CH_2_CH_2_CH_2_), 2.20–2.44 (m, 6H, CH_2_N(CH_2_)_2_), 3.20 (brs, 2H, -CH_2_-), 3.90 (brs, 4H, N(CH_2_)_2_), 6.52–8.31 (m, 11H, ArH). ^13^C NMR (300 MHz, CDCl_3_) *δ* (ppm): 170.56, 161.20, 157.82, 137.76, 133.87, 133.12, 131.84, 131.08, 128.14, 123.81, 117.36, 110.65, 55.13, 52.27, 42.15, 23.01. HRMS (ESI) calculated for C_26_H_26_BrN_5_O_4_S [M+H]^+^ 586.0944; found the following: 586.0867. Analysis calculated for C_26_H_26_BrN_5_O_4_S (584.48); C, 53.43; H, 4.48; N, 11.98; found the following: C, 53.72; H, 4.56; N, 11.86.

### 2.2. In Vitro Study

#### 2.2.1. Cyclooxygenase Inhibition Assay

To conduct the cyclooxygenase inhibition assay, we utilized a kit from the manufacturer that contained 150 μL of assay buffer, 10 μL of heme, and 10 μL of either COX-1 or COX-2 enzyme. Each sample was prepared in triplicate, with the addition of 10–500 μM of the test compounds and 10 μM of solvents such as methanol, ethanol, and dimethyl sulfoxide (DMSO). We then added 20 μL of N,N,N cent,N cent-tetramethyl-p-phenylenediamine (TMPD) to each well. Arachidonic acid was introduced to initiate the reaction, which was allowed to proceed for 2 min. Following the incubation period, the extent of TMPD oxidation was measured using the multiscan Go microplate reader at a wavelength of 590 nm (Thermo Fisher Scientific, Waltham, MA, USA).

#### 2.2.2. Biological Evaluation

##### Cell Line and Culture Conditions

NHDF (normal human dermal fibroblast) cells, sourced from the American Type Culture Collection (ATCC), were maintained in DMEM without phenol red, enhanced with 10% fetal bovine serum (FBS), 2 mM L-glutamine, and 25 μg/mL gentamicin. The cells were incubated under conditions of 5% CO_2_, at a temperature of 37 °C, and 95% humidity. We monitored the cells bi-weekly, utilizing or subculturing them once they reached over 70% confluence. For all experimental setups, cells were plated at a density of 10,000 cells per well and allowed to recover in a CO_2_ incubator for 24 h.

##### Tested Compounds

The compounds under investigation were dissolved in DMSO to achieve a final concentration of 10 mM and then stored at −20 °C. Lipopolysaccharide (LPS) from *Escherichia coli* (Sigma-Aldrich, Saint Louis, MO, USA; catalog no. L2630) was dissolved in distilled water to a concentration of 1 mM. The stock solution of LPS was stored at −20 °C for up to 6 months. Final concentrations of LPS used in this study were 50 µg/mL. Prior to use, the solutions were thawed at room temperature. For each compound, 8 different concentrations in the range of 500–10 μM were prepared in the culture medium.

##### MTT Assay

The cytotoxicity of the test compound was assessed using the MTT assay. After cell seeding and overnight incubation, the supernatant was removed, and the compounds were added at concentrations ranging from 500 to 10 µM. Control cells were treated with complete medium only, while cells treated with medium containing DMSO at the highest compound concentration were included as the solvent control. After the incubation period, the supernatant was removed, and 50 µL of a 1.0 mg/mL MTT solution was added to each well and incubated for 2 h at 37 °C. The medium was then removed, and 100 µL of DMSO was added to each well. Finally, absorbance was measured at 570 nm using a Multiskan GO spectrophotometer.

Cell viability was calculated according to Equation (1):
(Ai − Am)/(Ac − Am) × 100(Ai − Am)/(Ac − Am) × 100(1)
where Ai is the absorbance of the tested concentration, Am is the absorbance of the medium, and Ac is the absorbance of the control. Based on the cytotoxicity test results, the IC_50_ was calculated using a 4-point lognormal distribution with a Hill coefficient. Experiments were performed in triplicate.

Although the data showed a normal distribution, due to the lack of homogeneity of variance, Welch’s ANOVA was used for analysis. Post-hoc analysis was performed using Dunnett’s multiple comparison test to identify specific group differences. A significance level of *p* < 0.05 was used in all statistical tests. All statistical analyses of the MTT assay were performed using GraphPad Prism 8.0.2 software (GraphPad Software Inc., San Diego, CA, USA).

##### Inflammation Model

Cells seeded in 96-well plates were treated with 50 ug/mL LPS for 24 h for the MTT assay or 1 h for the DCF-DA assay. Then, the culture was washed and treated with the compounds with the most favorable COX-2/COX-1 ratio—**BS23** and **BS28** at concentrations of 10, 50, and 100 µM for the next 24 or 1 h. After this time, MTT and DCF-DA assays were performed according to the procedure described in Wiatrak et al. [22]

##### Gene Expression of TNF-α and IL-6 Detected by Real-Time RT-PCR

The NHDF cells were treated according to the previously outlined procedure. The control and LPS groups were incubated separately in culture dishes containing DMEM, while the test compounds groups were placed in dishes with 10 µM **BS23** and **BS28**, respectively. After a 24 h incubation at 37 °C in a 5% CO_2_ atmosphere, the NHDF cells were scraped from the bottom of the microplate using a rubber, and then they were centrifuged at 9000× *g* for 5 min. The pellet was washed with ice-cold PBS twice. RNA was isolated using a Total RNA Mini Plus kit (A&A Biotechnology, Gdańsk, Poland) followed by DNase I (Thermo Fisher Scientific, Waltham, MA, USA) treatment according to the producer’s instructions. cDNA synthesis was performed using a Fast Gene Scriptase II cDNA (Nippon Genetics Europe, Duren, Germany). RT-qPCR analyses were performed using the DyNAmo Flash SYBR Green qPCR Kit (Thermo Fisher Scientific, Waltham, MA, USA) and the CFX Connect Real-Time PCR Detection System (Bio-Rad, Hercules, CA, USA) . Relative gene expression was normalized to the reference gene, β-actin, using the 2^−ΔΔCT^ method. The primer sequences utilized in this study were as follows: *IL-6*: F, 5′-TGTCCCATGCCACTCAGAGA-3′; R, 5′-AGCAGGTGCTCCGGTTGTAT-3′; *TNF-α*: F, 5′-AGCCCATGTTGTAGCAAACC-3′; R, 5′-TGAGGTACAGGCCCTCTGAT-3′; *β-actin*: F, 5′-CTGTCTGGCGGCACCACCAT-3′; R, 5′-GCAACTAAGTCATAGTCCG-3′. All experiments were performed at least three times.

##### In Vitro Antioxidant Assays

2,2-Diphenyl-1-Picrylhydrazyl (DPPH) as an Antioxidant Capacity

The DPPH assay was conducted to assess the radical scavenging capabilities of the synthesized compounds. Stock solutions of the compounds were formulated in DMSO and subsequently diluted serially to achieve test concentrations ranging from 500 to 10 μM in ethanol. Each sample’s ethanolic DPPH solution was treated with the respective test concentrations and incubated at 37 °C. After a 30 min incubation period, the absorbance was measured at 517 nm, using ethanol as the blank control. Ascorbic acid served as the reference standard, and the IC_50_ values for the synthesized compounds were determined using the following Equation (2):(2)$$\%DPPH\;scavenging\;activity=\left(\frac{Abs\left(control\right)-Abs\;\left(sample\right)}{Abs\;(control)}\right)\times 100$$

ABTS Assay

The activity of ABTS (2,2′-azino-bis(3-ethylbenzothiazoline-6-sulphonic acid)) was assessed to evaluate the radical scavenging potential of the synthesized compounds. The ABTS reagent was generated by combining equal volumes of a 7 mM aqueous solution of ABTS and a 2.45 mM potassium persulfate solution. This mixture was then incubated in the dark at room temperature for 16 h. Following incubation, the solution was diluted with ethanol until the absorbance reached approximately 0.7 ± 0.05 at a wavelength of 734 nm. The stock solutions of the synthesized compounds in DMSO were serially diluted to obtain concentrations ranging from 500 μM to 10 μM in ethanol. The ABTS reagent was then treated with various concentrations of each compound and incubated in the dark at room temperature. After 30 min of incubation, the absorbance was measured at 734 nm, using ethanol as the blank control. Ascorbic acid served as the standard for comparison, and the IC_50_ values for the synthesized compounds were calculated using the following Equation (3):(3)$$\%ABTS\;scavenging\;activity=\left(\frac{Abs\left(control\right)-Abs\;\left(sample\right)}{Abs\;(control)}\right)\times 100$$

### 2.3. Differential Scanning Calorimetry (DSC)

Phospholipids: 1,2-dipalmitoyl-n-glycero-3-phosphatidylcholine (DPPC) and 1,2-dimyristoyl-*sn*-glycero-3-phosphatidylcholine (DMPC) were obtained from Merck KGaA (Darmstadt, Germany). Phospholipids were used as delivered without any purification. Tris-EDTA buffer solution was purchased from Merck KGaA (Darmstadt, Germany). Meloxicam was purchased in Alfa Aesar (Karlsruhe, Germany).For each calorimetric sample, 2 mg of phospholipid was dissolved in the appropriate amount of chloroform stock solution (5 mM) of the compound studied. The compound/lipid molar ratios in the samples were 0.06, 0.08, 0.10, and 0.12. Then, the solvent was evaporated by a stream of nitrogen, and the residual solvent was removed under a vacuum for 2 h (Rotary evaporator, Büchy Poland, Warsaw, Poland). Samples were hydrated by 20 μL of Tris–EDTA buffer (pH 7.4). Hydrated mixtures of phospholipid, compounds studied, and buffer closed in Eppendorf tubes were heated (Labnet Dry Bath, Labnet International Inc., Edison, NJ, USA) to a temperature higher by 10 °C than the main phase transition temperature of the phospholipid used and vortexed (neoVortex, neoLab) until homogeneous dispersion was obtained. Samples were sealed in aluminum pans type Concavus^®^ (Netzsch GmbH & Co., Selb, Germany). A pan of the same type, filled with 20 μL of Tris–EDTA buffer (pH 7.4), was employed as a reference. Calorimetric measurements were performed using a differential scanning calorimeter DSC 214 Polyma (Netzsch GmbH & Co., Selb, Germany) equipped with an Intracooler IC70 (Netzsch GmbH & Co., Selb, Germany) in the Laboratory of Elemental Analysis and Structural Research, Faculty of Pharmacy, Wroclaw Medical University. Measurements of the phospholipid’s main phase transition were performed using the heat-flow measurement method at a heating rate of 1 °C per minute over a temperature range of 30–50 °C in a nitrogen dynamic atmosphere (25 mL/min). Data were analyzed offline using Netzsch Proteus^®^ 7.1.0 (Netzsch GmbH & Co., Selb, Germany) analysis software. The transition enthalpies were stated in [J/g]. The measured heat was normalized per gram of lipid. The calorimeter was calibrated using standard samples from calibration set 6.239.2–91.3.00 supplied by Netzsch (Netzsch GmbH & Co., Selb, Germany). All samples were weighed on a Sartorius CPA225D-0CE analytical balance (Sartorius AG, Gottingen, Germany) with a resolution of 0.01 mg.

### 2.4. In Silico Study

#### 2.4.1. The Reactivity of Designed Compounds

The structures of arylpiperazine-1,2-benzothiazine derivatives were optimized at the B3LYP ++6-31G** level of theory in the Gaussian 2016 C.01 software package [23]. The solvent effects were included using the polarization continuum model (PCM) [24]. The reactivity parameters of compounds were calculated by the analogous method. The Avogadro molecular visualization program was used to visualize the calculated energies of frontier orbitals [25].

The reactivity of the molecules is related to the calculated energy values of the highest occupied molecular orbital (HOMO) and lowest unoccupied molecular orbital (LUMO) and their energy gap (Δɛ = ɛLUMO − ɛHOMO) [26,27]. Another parameter, the ionization potential (I) can be expressed as the negative energy of HOMO energy according to Koopman’s theory [28]. The electron affinity (A) was calculated as the negative value of LUMO energy. The methodology of calculation of the remaining global reactivity parameters is described in detail in the results section.

#### 2.4.2. Molecular Docking

In the present study, we used the standard docking protocol implemented in the AutoDock4.2 package to predict the binding mode of ligands to both cyclooxygenase isoforms (COX-1 and COX-2) [29]. The molecular targets downloaded from the Protein Data Bank were the structures of COX-1 (PDB ID:4O1Z) and COX-2 (PDB ID:4M11) co-crystalized with meloxicam [30]. The validation was performed by re-docking of reference drug into the protein crystal and comparing its position with the original structures. The root mean square deviation (RMSD) was estimated on the LigRMSD web server to evaluate the accuracy of docking prediction [31]. Due to the RMSD being found to be less than 2 Å, we assumed that the binding mode of compounds was correctly predicted. It should be underlined here that the scoring functions used in the molecular docking give approximate values of free energy of binding, so the computational results were validated by in vitro measurements. The preparation method of molecular target and inhibitor structures was presented in detail in our previous studies preparation [32,33]. According to previous research, the Lamarckian genetic algorithm is the most efficient and reliable approach of AutoDock4.2, so it was used with a total of 500 runs for each binding site [34]. The visualization of obtained results was performed in BIOVIA Discovery Studio visualizer and a Chimera package [35,36].

## 3. Results

### 3.1. Chemistry

The first stage of synthesis of final compounds **BS23**–**BS30** was the condensation of saccharin **1** with 2-bromoacetophenones substituted in the para position with a fluorine **2c** or bromine **2d** atom or a methyl **2b** or methoxy group **2a** (Figure 1). The next step was the *Gabriel–Colman* rearrangement reaction consisting of opening the 1,2-thiazole ring under the influence of sodium ethoxide at a temperature of 40–55 °C. In the second stage of this reaction, excess sodium ethoxide leads to ring closure by incorporating a carbon atom and forming a six-membered 1,2-thiazine ring. Subsequently, alkyl (**5** and **7**) and acyl (**6**) derivatives of arylpiperazine were obtained. Compounds **5** and **7** were obtained by alkylation of 1-(2-pyrido)piperazine or 1-(2-pyrimido)piperazine with 1-chloro-3-bromopropane, whereas compound **6** was obtained by reaction of 1-phenylpiperazine with chloroacetyl chloride. The last step towards obtaining the planned compounds **BS23**–**BS30** was the condensation of compounds **4a**–**d** with **5**, **6,** or **7** in anhydrous ethanol with the addition of sodium ethoxide (2.3%) at the boiling point of ethanol. The progress of the reaction was monitored by thin-layer chromatography (TLC). The obtained products were crystallized from ethanol and then submitted to spectroscopic analyses to confirm their structure and purity.

### 3.2. Evaluation of COX-1/COX-2 Selectivity and Potency

#### 3.2.1. In Vitro Cyclooxygenase Inhibition Assay

Cyclooxygenase (COX) peroxidase activity was evaluated using a colorimetric assay. The impact of meloxicam and the analyzed compounds on the activities of both cyclooxygenase isoforms (COX-1 and COX-2) was measured. The incubation period was maintained at 2 min, following the manufacturer’s guidelines. Considering the well-established side effects of nonsteroidal anti-inflammatory drugs (NSAIDs), such as aspirin, naproxen, and ketoprofen, which typically inhibit COX-1, our goal was to compare the effects of newly synthesized compounds with meloxicam. Meloxicam inhibits both isoforms but has a higher affinity for COX-2. We determined the concentrations of all compounds required for 50% inhibition of COX-1 and COX-2. IC_50_ values were then calculated, and the ratio of IC_50_ values for both cyclooxygenase isoforms allowed us to assess the selectivity of the tested structures for COX-1 and COX-2. The full set of IC_50_ values for both enzymes and the selectivity coefficient are shown in Table 1.

In the cyclooxygenase inhibition assay, **BS** compounds exhibited varying degrees of efficacy against COX-1 and COX-2 isoforms, with selectivity ratios (COX-2/COX-1) indicating a relative preference for COX-2 over COX-1.

#### 3.2.2. Cytotoxicity Assay

Meloxicam demonstrated an IC_50_ value of 267.71 ± 8.1 µM for COX-1 and 112.67 ± 3.3 µM for COX-2, resulting in a selectivity ratio of 0.42. This indicates that meloxicam has a moderate preference for inhibiting COX-2 over COX-1. The IC_50_ value for meloxicam in the MTT assay was 174.23 ± 20.3 µM.

Among the **BS** compounds, **BS23** exhibited an IC_50_ value of 241.64 ± 4.2 µM for COX-1 and 13.19 ± 2.1 µM for COX-2, with a selectivity ratio of 0.05. This compound showed a significantly higher selectivity for COX-2 compared to meloxicam. Additionally, the IC_50_ value in the MTT assay was 228.40 ± 28.1 µM, indicating lower cytotoxicity compared to meloxicam. **BS24** had an IC_50_ value of 126.54 ± 3.6 µM for COX-1 and 26.34 ± 3.4 µM for COX-2, giving a selectivity ratio of 0.21. This selectivity is still more pronounced for COX-2 than for meloxicam, with an IC_50_ value in the MTT assay of 163.46 ± 15.7 µM, indicating slightly higher cytotoxicity. **BS25** had IC_50_ values of 130.96 ± 7.5 µM for COX-1 and 62.63 ± 1.7 µM for COX-2, with a selectivity ratio of 0.56. This compound showed a lower selectivity for COX-2 compared to meloxicam, with an IC_50_ value in the MTT assay of 200.10 ± 20.9 µM. **BS26** showed IC_50_ values of 128.73 ± 9.1 µM for COX-1 and 18.20 ± 4.3 µM for COX-2, with a selectivity ratio of 0.14, indicating greater selectivity for COX-2 than meloxicam. Additionally, this compound showed lower cytotoxicity compared to meloxicam, with an IC_50_ value of 203.28 ± 22.9 µM. **BS27** had IC_50_ values of 116.89 ± 4.3 µM for COX-1 and 25.55 ± 2.0 µM for COX-2, with a selectivity ratio of 0.22, also indicating greater selectivity for COX-2 than meloxicam. Its MTT assay IC_50_ was 190.60 ± 12.5 µM. **BS28** showed IC_50_ values of 95.88 ± 5.7 µM for COX-1 and 12.46 ± 1.9 µM for COX-2, with a selectivity ratio of 0.13, indicating a strong preference for COX-2 over COX-1 and better selectivity than meloxicam. Its MTT assay IC_50_ was 179.74 ± 13.6 µM. **BS29** presented IC_50_ values of 124.81 ± 6.7 µM for COX-1 and 17.80 ± 2.1 µM for COX-2, with a selectivity ratio of 0.14, indicating higher selectivity for COX-2 compared to meloxicam. Its MTT assay IC_50_ was 232.51 ± 11.8 µM. **BS30** had IC_50_ values of 121.51 ± 3.4 µM for COX-1 and 59.22 ± 3.7 µM for COX-2, with a selectivity ratio of 0.49, which is comparable to meloxicam. Its MTT assay IC_50_ was 168.50 ± 30.4 µM, indicating slightly higher cytotoxicity.

In summary, many **BS** compounds showed higher selectivity for COX-2 over COX-1 compared to meloxicam, with **BS23, BS26, BS28**, and **BS29** demonstrating particularly strong selectivity for COX-2. This suggests that these compounds may have potential advantages as anti-inflammatory agents with fewer side effects associated with COX-1 inhibition.

The simplification by removing the nitrogen atom from the aromatic rings appears to be beneficial. These modifications significantly increase the selectivity for COX-2. For example, **BS23**, in which piperazine has a pyridine ring instead of pyrimidine, shows a strong preference for COX-2 with a selectivity coefficient of 0.05 compared to 0.42 for meloxicam. This suggests that removing the nitrogen atom may reduce interactions with COX-1, thereby increasing the selectivity for COX-2.

Additionally, introducing halogen atoms (e.g., fluorine and bromine) into the structure may affect the binding affinity for COX enzymes. For instance, **BS26**, **BS28**, and **BS29**, which contain a fluorine or bromine atom, show selectivity coefficients of 0.14, 0.13, and 0.14, respectively, indicating an increased selectivity for COX-2.

The structural modifications in **BS** compounds highlight the delicate balance between achieving high COX-2 selectivity and maintaining low cytotoxicity. Removing specific atoms, simplifying ring structures, and introducing or removing functional groups can significantly impact the binding affinity and selectivity toward COX enzymes.

### 3.3. The Interactions with Phospholipid Bilayers

Cyclooxygenase is a membrane protein, associated with the phospholipid bilayer surrounding the endoplasmic reticulum, as well as the cell nucleus, which is why it is so important to investigate the interaction of new compounds with model membranes.

In the present work, we have used differential scanning calorimetry (DSC) to determine the interactions of studied compounds with phospholipid bilayers as artificial models of biological membranes.

The impact of two studied compounds on the lipid thermal behavior is presented in Figure 3, showing the example thermograms of DPPC mixed with **BS23** and **BS24** at different molar ratios. The addition of the compounds caused the disappearance of the DPPC pretransition and concentration-dependent shift of the main transition temperature towards lower values, accompanied by a decrease in the transition peaks area and the broadening of the peaks. Despite the slight difference in the chemical structure of both compounds (pyrimidine instead of pyridine), the effects are more pronounced for **BS24** (Figure 3B) than for **BS23** (Figure 3A).

All examined compounds decreased the main transition temperature of DPPC in a concentration-dependent manner (Figure 4A). The addition of studied compounds to DPPC also resulted in the broadening of the transition peaks. Moreover, all examined compounds decreased the enthalpy of the DPPC main phase transition. In the case of all DPPC gel–liquid crystalline phase transition parameters, the most pronounced effects were found for compounds **BS24**, **BS23**, and **BS30**.

Compounds **BS26** and **BS27** had the smallest impact on the transition temperature. The **BS27** compound with the highest molar ratio (0.12) lowered the temperature of the main phase transition to 39.2 °C, which is a reduction in the transition temperature of the DPPC phospholipid by only 2 °C. The **BS27** compound at the same concentration reduced the temperature to 39.075 °C, i.e., by 2.13 °C compared to the pure DPPC phospholipid. The greatest impact on the main transition temperature change was observed for the compounds **BS23**, **BS24**, and **BS30**. For example, for the **BS24** compound in the highest test compound: the lipid ratio, the change in the temperature of the main phase transformation in relation to the pure DPPC phospholipid, was 4.425 °C. All tested compounds had a more pronounced effect on model phospholipid membranes than meloxicam (Figure 4).

Figure 5 shows the influence of studied compounds on the DPPC peak half-width of the DPPC main phase transition. Each compound tested caused an increase in the half-width of the main phase transition peak in relation to the pure DPPC. The following compounds had the smallest impact on this parameter: **BS25**, **BS26**, and **BS27**. For example, **BS25** and **BS27** at the highest molar ratios broadened the main phase transition peak by only 1.25. The greatest increase in the half-width of the DPPC main phase transition was caused by **BS24** and **BS30**. (The changes in the DPPC peak half-width in relation to the pure phospholipid for these derivatives were 2 and 2.125).

All examined compounds decreased the main transition temperature of DPPC and DMPC, so they perturbed the phospholipid multi-bilayer structure. The stronger change in this parameter was observed in the presence of DMPC (Figure 6 open symbols) in comparison to DPPC (Figure 6 full symbols). The greatest reduction in Tm was demonstrated by the following compounds: **BS23**, **BS25**, and **BS30** in comparison to DMPC. The smallest effect on the main transition temperature was observed for compounds **BS26** and **BS27** for DPPC. DMPC is a phospholipid with a shorter acyl chain than DPPC. The weaker interactions between hydrocarbon chains and the more loosely packed structure of DMPC in comparison to DPPC, could provoke easier compound incorporation into bilayers formed from DMPC.

### 3.4. Anti-Inflammatory Properties of Designed Compounds

#### 3.4.1. Inflammation Assay

NHDF cell cultures were treated with LPS at a concentration of 50 µg/mL to induce pro-inflammatory cytokines such as IL-1β, TNF-α, and IL-6 through the activation of the TLR4 pathway [37]. Subsequently, the cells were treated with selected compounds **BS23** and **BS28** due to their favorable COX2/COX1 ratio. The results are presented in Figure 7A. Within the tested concentration range, the cell viability after treatment with **BS23** was around 70% of the value for untreated (control) cells. For **BS28** at the same concentrations, the absorbance values ranged between 0.64 and 0.68. Additionally, the absorbance value for LPS was 0.56. Compared to the LPS control, both compound samples showed higher cell viabilities, suggesting that these compounds may reduce LPS-induced cytotoxicity. The **BS23** compound exhibited a slightly stronger protective effect than **BS28**.

The results presented in Figure 7B pertain to the ROS (reactive oxygen species) test using DCF-DA, conducted on NHDF cells incubated with LPS at a concentration of 50 µg/mL for 1 h, followed by treatment with compounds **BS23** and **BS28** for an additional hour. For both tested compounds, **BS23** and **BS28**, within the examined concentration range, the ROS values were similar to the ROS levels in the control culture (cells incubated in medium only). These values were significantly lower than those for the LPS control, where the ROS level was nearly twice as high compared to the control (medium culture), with a value of 1.76.

#### 3.4.2. Effect of New Meloxicam Derivatives on the Cytokine mRNAs Level in Inflamed Cells

After a 24 h incubation period of inflamed normal human dermal fibroblast (NHDF) cells with meloxicam derivatives, specifically **BS23** and **BS28**, the expression of pro-inflammatory cytokines TNF-α and IL-6 was assessed through real-time RT-PCR analysis (Figure 8). Compared to the control group, the lipopolysaccharide (LPS)-treated cells exhibited a significant upregulation of TNF-α (Figure 8 left panel) and IL-6 (Figure 8 right panel), demonstrating that LPS stimulation effectively induces the expression of genes associated with inflammatory responses. Conversely, treatment with meloxicam derivatives (**BS23** and **BS28**) markedly reduced the expression levels of TNF-α and IL-6 in comparison to the LPS group. These findings suggest that **BS23** and **BS28** may suppress inflammation by inhibiting the gene expression of key pro-inflammatory cytokines in inflamed NHDF cells.

The observed anti-inflammatory effects of **BS23** and **BS28** align with previous research on nonsteroidal anti-inflammatory drug (NSAID) derivatives, which have been shown to mitigate the inflammatory response by targeting pathways that regulate cytokine production. Studies suggest that TNF-α and IL-6 play pivotal roles in mediating chronic inflammation and are often used as biomarkers to assess the efficacy of anti-inflammatory treatments [38]. The significant downregulation of these cytokines following treatment with meloxicam derivatives highlights their potential therapeutic value. Similar outcomes have been reported with other NSAID analogs, which also demonstrate the capacity to inhibit pro-inflammatory gene expression, thereby reducing the overall inflammatory response [39]. Further exploration into the molecular mechanisms underlying the action of **BS23** and **BS28** could provide valuable insights into their clinical applications.

#### 3.4.3. Antioxidant Assays

The antioxidant properties of meloxicam and its synthesized derivatives (**BS23**–**BS30**) were evaluated using two different in vitro assays, reflecting the common practice of applying methods to assess the antioxidant capacity of a sample [40]. The + radical scavenging of these compounds was summarized using IC_50_ values (Table 2).

Commonly employed assays like DPPH and ABTS are widely recognized for assessing the radical scavenging potential of compounds by measuring their capacity to neutralize free radicals or donate hydrogen atoms, thereby determining antioxidant activity [41]. The ABTS assay is considered superior to DPPH due to its solubility in both water and organic solvents and its quicker reaction rate with the test compounds [42]. The interaction of the tested compounds with the stable DPPH radical demonstrated their free radical scavenging ability, with this interaction being concentration dependent. However, the IC_50_ values for meloxicam and its derivatives in the DPPH assay exceeded 500 μM. In contrast, in the ABTS radical scavenging assay, the meloxicam derivatives displayed a notably higher scavenging potential in a concentration-dependent manner compared to both meloxicam and ascorbic acid (standard antioxidant). While meloxicam is known to possess inherent antioxidant properties and has been reported to reduce reactive oxidant formation [43], its derivatives like **BS23** and **BS28** significantly improved its ABTS radical scavenging capacity.

### 3.5. The Reactivity of Designed Compounds

It is well known that frontier molecular orbital energies can be used to determine various reactivity parameters of compounds. The ability of molecules to donate electrons is indicated by the high value of HOMO energy. The compounds with a low value of LUMO energy are capable of accepting electrons. Therefore, the visualization of these orbital features enables the identification of molecular regions responsible for interactions in the molecular target pocket. As presented in Table 3 and in the Appendix A, the regions of high electron density in the case of pyrimidinylpiperazine derivatives (**BS24**, **BS30**) are localized on 1,2-benzotiazine fragment of molecules. Such a distribution of electron density may indicate the region of compounds capable of scavenging free radicals as well as the area participating in charge transfer, hydrophobic, and π-stacking interactions. In the cases of compounds **BS23**, **BS25**, **BS26**, **BS28,** and **BS29,** the region of high electron density is localized on phenylpiperazine and pyridinylpiperazine fragments. In all considered compounds, LUMO orbitals existed over a phenyl ring with various alkyl substituents.

It should be also mentioned that the HOMO-LUMO energy gap determines the stability and reactivity of the molecules. Namely, the larger energy gap indicates the lower reactivity and the greater stability of molecular systems. According to the data presented in Table 3, compound **BS24** has the largest value of Δε (3.600 eV), which suggests its lower reactivity and greater stability. Compound **BS29** is characterized by the lower value of the gap. It indicates higher reactivity of the system. The energy of HOMO orbitals of the molecules corresponds with their ionization potential (I). Due to the lowest value of the I compound, **BS28** shows better donor electron properties than other derivatives. On the other hand, the largest values of electron affinity of **BS29** and **BS30** suggest their ability to accept electrons.
$$A=-E_{\mathrm{LUMO}};\;I=-E_{\mathrm{HOMO}};\;\eta =\frac{I-A}{2};\;s=\frac{1}{2\eta };\;\mu =\frac{-\left(I+A\right)}{2};\;\chi =\frac{\left(I+A\right)}{2};\;\omega =\frac{\mu^{2}}{2\eta }$$

The next parameter, the hardness of molecules (η), can be calculated following Expression (4):(4)$$\eta =\frac{I-A}{2}$$
based on Janak’s theorem [44]. The second related parameter, namely softness (s), is calculated from Equation (5).
(5)$$s=\frac{1}{2\eta }$$

Both could be useful in determining the intermolecular charge transfer phenomenon of molecular systems. It is known that the hardness increases while the polarizability of molecules decreases. Additionally, η and s determine the stability and reactivity of compounds.

Compound **BS24** is characterized by the highest value of hardness (4.726), and compound **BS28** is the softest molecule (0.116).

Another parameter based on the thermodynamic properties of the molecules is the global electrophilicity index (ω) expressed by Equation (6) [45].
(6)$$\omega =\frac{\mu^{2}}{2\eta }$$

The electrophilicity index is defined as the change of energy due to the flow of electrons between the donor and acceptor groups of molecules. Studies have shown an existing correlation between ω and the biological activity of compounds [46]. It is clear from Table 4 that compounds **BS28**, **BS29,** and **BS30** are characterized by the highest values of electrophilicity indexes and can therefore be the most biologically active.

Very important parameters are the chemical potential of the molecule (*μ*) and its electronegativity (*χ*). The first one describes the charge transfer within a system in the ground state and can be expressed as follows (see Equation (7)).
(7)$$\mu =\frac{-\left(I+A\right)}{2}$$

Compound **BS30** has the greatest value of chemical potential, so it is the most reactive of all derivatives. Electronegativity measures the attraction of an atom for electrons in a covalent bond numerically; it can be expressed with Equation (8).
(8)$$\chi =\frac{\left(I+A\right)}{2}$$

According to the calculations, compound **BS30** had a higher value χ and better electron-attracting capability than others.

### 3.6. Molecular Docking Studies

According to the scoring function including free enthalpy of binding, all designed compounds show a slightly stronger binding affinity for COX-2 (see Table in the Appendix A). The data are in agreement with this experimental study. The value of ΔG_binding_, which can characterize the ability of compounds to bind to the active site of protein, varies from −15.1 kcal/mol to −15.6 kcal/mol (Table 5).

As presented, the potency of binding to the COX-2 active site in all considered cases is almost the same. The structural modifications of compounds influence insignificantly their binding mode. All designed inhibitors can bind to the specific binding pocket arising in the structure of COX-2 due to the presence of Val523 instead of Ile523 and the change of Tyr355 conformation in the structure of COX-2 [47]. The specific cavity is created by Leu352, Ser353, Tyr355, Phe518, and Val523. As presented in Figure 9, compound **BS23** can form one hydrogen bond with Ser530 as a classic anti-inflammatory agent. The pyridinylpiperazine moiety binds to COX-2 through π-type and van der Waals interactions with Leu384, Met522, Val349, Val523, Gly526, and Ala527 and Phe381, Tyr385, and Trp387, respectively. As determined, the benzothiazine scaffold is responsible mainly for van der Waals interactions with Met113, Leu117, Ser353, and π-type forces with Ile345, Leu531, and Met535 amino acid residues. A similar mode of binding exhibits remaining derivatives (see Appendix A).

The data obtained for the COX-1 in complex with designed inhibitors are presented in detail in the Appendix A. 

## 4. Discussion

Our study presents a comprehensive analysis of synthesized 1,2-benzothiazine derivatives in terms of their potential for anti-inflammatory and antioxidant activity. The results revealed the selective activity of compounds towards COX-2, with some compounds showing significant COX-2 preference compared to meloxicam. The selectivity is advantageous for reducing side effects, aligning with literature on selective COX-2 inhibitors, which suggests their reduced risk of gastrointestinal adverse effects common in COX-1 inhibitors [48,49,50]. As presented in Table 1, the most selective compound is **BS23;** however, compound **BS24**, **BS25**, **BS27**, **BS28,** and **BS29** show a better selectivity profile than meloxicam. Only two of the proposed derivatives have selectivity close to that of the reference drug (selectivity ratio: (COX-2/COX-1) about 0.40). On the other hand, only compounds **BS24** and **BS30** show greater toxicity than meloxicam. Many studies discussed the structural factors determining for COX-2 selectivity of NSAID derivatives, reinforcing the current study’s approach. The binding site of COX-2 is predominantly hydrophobic, facilitating interactions with lipophilic substrates and inhibitors. The presence of polar amino acid residues such as Arg120 and Ser530 allows hydrogen bonding, which is critical for selectivity. In addition, the side pocket of COX-2 is more flexible than that of COX-1, allowing the accommodation of bulkier substituents in selective inhibitors. According to molecular docking proposed compounds similar to classic nonsteroidal anti-inflammatory drugs (diclofenac) can form hydrogen bonds with Arg120, Tyr355, and Ser530 [51]. As presented in Table 5, compounds **BS23** and **BS30** form two hydrogen bonds with arginine and serine amino acid residues of COX-2. On the other hand, similar to meloxicam, all compounds occupy the hydrophobic pocket and are surrounded by Ser353, Leu384, Tyr385, Trp387, Val523, and Met522, and the differences in free energy of binding are negligible [47].

The selectivity of inhibitors with various structural modifications was analyzed by Gautam et al. [47]. They proved that specific substitutions, such as halogens and nitrogen-based groups, enhance COX-2 binding affinity while reducing interactions with COX-1. There are also some studies on benzo[d]thiazol derivatives that indicate a significant role of halogens as electron-withdrawing substituents in determining anti-inflammatory properties providing the best orientation within the active site of COX-2 [52]. This structural approach is reflected in the results of our study, particularly with compounds like **BS26**, **BS28**, and **BS29**, where halogen groups improve COX-2 selectivity. Another study on oxicam analogs revealed that slight changes in molecular structure can significantly impact COX binding [30]. As can be seen from Table 1, removing nitrogen atoms from aromatic substituent appears to be important for selectivity (compound **BS28**). The replacement of the methoxy group with a methyl substituent in the **BS24** structure results in reduced selectivity. As we know, the methoxy group is a strong electron donor substituent. When attached to an aromatic ring, it increases the electron density of the ring, which can improve π–π stacking interactions with the hydrophobic regions of the COX-2 active site. As concluded in the earlier studies on pyrazole analogs with methoxy substituents, these groups contribute additionally hydrogen bonding with residues like Lys68, Tyr108, and Arg106, increasing binding energy and improving COX-2 inhibition compared to the selective inhibitor celecoxib. Figure 9 and Appendix A clearly show that compounds with the methoxybenzene moiety (**BS23** and **BS24**) promote π-type interactions with amino acids of the protein. As presented, the methoxybenzene ring of compound **BS23** can interact with Val116 and Arg120 via π–σ and π–cation interactions, respectively. In the case of compound **BS24,** π–alkyl and π–π T shape interactions with Val116 and Tyr355 localized specifically for the COX-2 cavity are possible.

Drugs that are COX inhibitors should also interact with cell membranes. The natural substrate for COX is arachidonic acid, which occurs among the phospholipids of the cell membrane. COX is a membrane-bound protein of the endoplasmic reticulum. Potential COX inhibitors should have some ability to interact with the cell membrane to enter the COX channel containing its active site. A potential NSAID candidate must therefore demonstrate an appropriate level of lipophilicity and interaction with the cell membrane. Additionally, interaction with the model membrane will therefore be more informative about whether the compound can penetrate membranes, be absorbed from the gastrointestinal tract, and have an affinity for the membrane. The results of the DSC measurements show that the studied compounds can interact with the phospholipid phase of the membrane. In the case of all phospholipid gel–liquid crystalline phase transition parameters, the most pronounced effects were found for compounds **BS23**, **BS24,** and **BS30**. The vanishing of phospholipid pretransition, broadening of transition peaks, and decrease in the main transition temperature of the phospholipid may suggest that the studied compounds interact with the phospholipid head groups, changing lipid packing, probably by increasing the spacing between lipid heads, which may result in disruption in the rippled phase of the DPPC. Zambrano et al. published a revision of using the differential scanning calorimetry method in investigating drug–membrane interactions. They concluded that the DSC method enables understanding the impact of the studied compounds on lipid packing and stability of the phospholipid bilayer [52]. There are some data concerning selected NSAIDs analyzed by Lichtenberger et al. using a model membrane made of DPPC. The obtained results indicated that these drugs interact with non-charged phospholipids contained in the mucosa of the stomach and intestines by forming complexes with them. This interaction may promote the gastrotoxic effect of those medicines [53]. Lucio et al., on the other hand, showed that acemetacin, indomethacin, and nimesulide influenced the phase transition temperature of the model membrane. It was concluded that this effect may have a negative impact on the gastrointestinal mucosa, which is associated with the occurrence of side effects of these medicines [54]. The thermal effects of selected oxicam derivatives were also evaluated by Kyrikou et al. using the DSC method. Their results were similar to ours. The perturbing effect of the drugs exerted on DPPC model membranes resulted in a decrease in the main phase transition temperature and in the broadening of the transition peaks [55].

As presented in the *Results* section, the anti-inflammatory potential was further supported by in vitro assays, showing that both **BS23** and **BS28** decreased levels of TNF-α and IL-6, cytokines crucial to inflammation pathways in lipopolysaccharide (LPS)-stimulated cells. Additionally, both derivatives visibly reduce the expression levels of pro-inflammatory cytokines in comparison to the LPS group. Hence, compounds can inhibit the gene expression of TNF-α and IL-6 in inflamed cells. This is consistent with reports on other NSAID analogs. Shabbir et al. examined 3-ethoksyhydroxyphenyl derivatives of benzothiazines (EHP). They showed that treatment of EHP significantly reduces pro-inflammatory tissue necrosis factor-α and interleukin-17 [56]. Our results demonstrated a robust correlation between the compounds’ COX inhibitory activity and their antioxidant properties. Several compounds exhibited promising anti-inflammatory activity and favorable reactivity profiles including the capacity to neutralize ROS. Additionally, analyzed derivatives (**BS28** and **BS29**) are characterized by the high value of electrophilicity indexes, which is a potential predictor of inflammatory activity. Compound **BS28** demonstrated the lowest value of the HOMO-LUMO gap indicating a stronger tendency to react with free radicals. The antioxidant capacity of proposed compounds also aligns with an earlier study. Erturk et al. proved that NSAID derivatives with strong radical scavenging properties not only reduce inflammation but also mitigate oxidative stress associated with chronic inflammatory conditions [43]. The analysis of structurally similar compounds showed the correlation between the reactivity of thiazine derivatives and their biological properties [57].

In summary, the following structural elements should be considered when designing selective COX-2 inhibitors:The substitution of Ile523 in COX-1 with Val523 in COX-2 creates space for selective inhibitors with bulky hydrophobic groups; therefore, compounds with the specific side chains that exploit this pocket should be designed (e.g., 1-(2-pyrido)piperazine fragment of our compounds);The sulfonamide group able to interact with polar amino acid residues as arginine or serine via hydrogen bonds (**BS23**, **BS24**, **BS26**, **BS27**, and **BS30**);Halogen atoms enhance binding affinity and selectivity by improving hydrophobic and dipole interactions within the COX-2 pocket;Aromatic scaffolds enhance π–π stacking interactions with hydrophobic residues in COX-2;Methoxy substituent, which contributes to increased hydrophobic interactions within the binding pocket and enhances π–π stacking with aromatic residues in the COX-2 active site (compound **BS23** and **BS24**).

## 5. Conclusions

Our findings suggest that the 1,2-benzothiazine derivatives could serve as potential lead candidates for the development of novel anti-inflammatory agents. They could induce a dual therapeutic effect, both inhibiting COX-2 and reducing ROS levels, potentially improving therapeutic outcomes in oxidative stress-driven inflammatory diseases. The data revealed that all compounds exhibit high or moderate selectivity towards the cyclooxygenase 2 isoform and low cytotoxicity in comparison to the reference drug. It is certain, however, that further in vivo studies are warranted to validate their therapeutic efficacy and safety.

## Data Availability

Data are contained within the article and Appendix A.

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
