# Peer review of "Anti-Inflammatory Properties of Novel 1,2-Benzothiazine Derivatives and Their Interaction with Phospholipid Model Membranes"

_membranes, 2024, doi:10.3390/membranes14120274_

Round 1

Reviewer 1 Report

Comments and Suggestions for Authors

This manuscript is devoted to anti-inflammatory properties of novel 1,2-benzothiazine derivatives and their interaction with phospholipid model membranes. A series of 1,2-benzothiazine derivatives were synthesized. Their interactions with lipid bilayers, ability to inhibit COX-1 and COX-2 activity, antioxidant activity were investigated. The work is of interest because the results indicate that the 1,2-benzothiazine derivatives could serve as potential lead candidates for the development of novel anti-inflammatory agents. I think that this manuscript may be published after minor revision.

Notes:

1. Authors should avoid any abbreviations in the Abstract of this manuscript.

2. Why the yields of desirable products BS23 – BS30 are so low (25-50%)? What is represent the residue?

3. The Scheme synthesis of 1,2-benzothiazine derivatives BS23 - BS30 (Scheme 1) should be increased for clarity. Reaction conditions and radicals are written too small. In the scheme the yields of products should be presented for clarity.

4. There are some printing mistakes that should be checked and corrected. For example, in the lines 503-504 the phrase “in Figure 2” should be changed by “in Figure 3”. In the line 501 “fosfolipid” should be changed by “phospholipid”.

5. It was demonstrated that all examined compounds decreased the main transition temperature of DPPC and DMPC. As a results how do these compounds act on the membrane? Do they loosen or compact the phospholipid bilayers? It should be noted more detailed. What does the decrease in the main transition temperature of model membrane indicate?

Author Response

Reviewer 1#

This manuscript is devoted to anti-inflammatory properties of novel 1,2-benzothiazine derivatives and their interaction with phospholipid model membranes. A series of 1,2-benzothiazine derivatives were synthesized. Their interactions with lipid bilayers, ability to inhibit COX-1 and COX-2 activity, antioxidant activity were investigated. The work is of interest because the results indicate that the 1,2-benzothiazine derivatives could serve as potential lead candidates for the development of novel anti-inflammatory agents. I think that this manuscript may be published after minor revision.

We are pleased that Reviewer found our paper interesting and appreciated the quality of

our results. Present paper has been revised according to Reviewer comments. Below is

the list of changes introduced to the present version of the manuscript:

Notes:

  1. Authors should avoid any abbreviations in the Abstract of this manuscript.

We have removed abbreviations from the abstract.

  1. Why the yields of desirable products BS23 – BS30 are so low (25-50%)? What is represent the residue?

During synthesis, many things contribute to the formation of a smaller amount of product than the theoretical yield suggests. Reactions in organic chemistry rarely proceed as planned on paper in reality, because they usually do not proceed in a single path. As a rule, in addition to the planned reaction pathway, other pathways leading to other products are created, which consume the substrates and cause the planned product to not be formed with 100% yield. This is a common problem both in small laboratories and in industry. Therefore, a number of measures are taken to increase the yield of the reaction, such as using a catalyst, changing the temperature or pressure during the reaction. An additional reason for the loss of product is the purification or isolation process. Sometimes it is better to decide to lose 10% of the product during additional purification, because obtaining a pure product is much more important than obtaining a larger amount of a more impure product. In this work, no attempts were made to optimize the reaction, so the yield of the planned products is not very high. However, for organic reactions, this is not a low yield either.

  1. The Scheme synthesis of 1,2-benzothiazine derivatives BS23 - BS30 (Scheme 1) should be increased for clarity. Reaction conditions and radicals are written too small. In the scheme the yields of products should be presented for clarity.

Thank you for this valuable comment. We have complied with the reviewer's comment. The scheme has been prepared as suggested.

  1. There are some printing mistakes that should be checked and corrected. For example, in the lines 503-504 the phrase “in Figure 2” should be changed by “in Figure 3”. In the line 501 “fosfolipid” should be changed by “phospholipid”.

We have made the corrections.

  1. It was demonstrated that all examined compounds decreased the main transition temperature of DPPC and DMPC. As a results how do these compounds act on the membrane? Do they loosen or compact the phospholipid bilayers? It should be noted more detailed. What does the decrease in the main transition temperature of model membrane indicate?

The vanishing of phospholipid pre-transition, broadening of transition peaks and decrease of the main transition temperature of the phospholipid may suggest that studied compounds interact with the phospholipid head groups, changing lipid packing, probably by increasing the spacing between lipid heads what may result in disruption in rippled phase of the DPPC. The results of DSC measurements show that the studied compounds can interact with the phospholipid phase of the membrane. Drugs that are COX inhibitors should also interact with cell membrane. The natural substrate for COX is arachidonic acid, which occurs among the phospholipids of the cell membrane. COX is a membrane-bound protein of the endoplasmic reticulum. Potential COX inhibitors should have some ability to interact with the cell membrane to enter the COX channel containing its active site. A potential NSAID candidate must therefore demonstrate an appropriate level of lipophilicity and interaction with the cell membrane. However, interaction with the lipid phase of the membrane is a different issue than selectivity for COX and good binding with the enzyme itself. Interaction with the model membrane will therefore be more informative about whether the compound will penetrate membranes, be absorbed from the gastrointestinal tract, and have an affinity for the membrane in which the COX protein is anchored. We also explained this in the discussion section.

Reviewer 2 Report

Comments and Suggestions for Authors

Despite overall positive impression a few points must be addressed:

Major point and study questions: Almost no correlation discussion is present and discussion is very underwhelming.

For example, discussing correlation between phase transition temperature effect and COX inhibition activity, or correlation with other parameters would make a great argument to highlight best performing molecules. Furthermore, some principles for future design of anti inflammatory agents could be extracted from this analysis and are very welcome to be discussed in the paper.

Very little comparison is done with the literature examples in the discussion. Please provide comparisons with current work on anti-inflammatory drug design, preferably multiple examples and with similar research methods such as phase transition analysis and COX inhibition studies. 

Same is true for introduction, very little current literature is discussed. Examples of novel anti-inflammatory compounds should be provided, and their pros and cons compared to the studied molecules should be discussed.

This would significantly improve paper quality.

What is a possible explanation of why compounds BS23 and BS30 more significantly affect lipid phase transition temperature, and demonstrate more beneficial docking energy (compared to other studied compounds), while the COX docking site is clearly in the middle of the protein, where direct interaction with lipid molecules is unlikely?

Could there be correlation between the type of docking interactions demonstrated for each compound with its performance in inhibition of COX-2 or lowering of phase transition temperature? For example, Compound BS30, which is among the best in the study demostrated the least amount of van der Waals interactions and most amount of pi-type interactions. Is it related to the hydrophobicity of molecule moieties and type of incorporated functional groups? What could be said about poorly performing molecules, what are they lacking?

Few minor points:

Figure 6 is difficult to comprehend. What exactly is behind the term "relative transition temperature"? Is it a relative value? Why is the Y-axis scale in absolute degrees then?

Why are error bars so tall? Error bars on figures 4,5 where, supposedly, parent data is shown, seem to be smaller. 

Also the Y-scale is chosen poorly, showing very little difference between bars.

Page 6 row 234, what is TMPD? 

On somefigures text should be enlarged 2-3 times. For example, on Scheme 1, text is very small and hard to read, but there is plenty of free space present. Consider text on other figures too.

Some errors and typos: page 12 row 501 "fosfolipid"; page 21 row 779 and page 22 row 787: leftover strikethrough formatting.

Author Response

Reviewer 2#

We thank the reviewer for his critical comments. The present paper has been revised according to all Reviewer comments. Below is the list of changes introduced to the present version of the manuscript:

Major point and study questions: Almost no correlation discussion is present and discussion is very underwhelming.

For example, discussing correlation between phase transition temperature effect and COX inhibition activity, or correlation with other parameters would make a great argument to highlight best performing molecules. Furthermore, some principles for future design of anti inflammatory agents could be extracted from this analysis and are very welcome to be discussed in the paper.

The vanishing of phospholipid pre-transition, broadening of transition peaks and decrease of the main transition temperature of the phospholipid may suggest that studied compounds interact with the phospholipid head groups, changing lipid packing, probably by increasing the spacing between lipid heads what may result in disruption in rippled phase of the DPPC. The results of DSC measurements show that the studied compounds can interact with the phospholipid phase of the membrane.  However, interaction with the lipid phase of the membrane is a different issue than selectivity for COX and good binding with the enzyme itself. Interaction with the model membrane will therefore be more informative about whether the compound will penetrate membranes, be absorbed from the gastrointestinal tract, and have an affinity for the membrane in which the COX protein is anchored.  A potential drug candidate may therefore interact better with the lipid phase of the membrane, so it will probably be better absorbed from oral administration and more easily achieve the goal of action, which is the COX protein, which is bound to the cell reticulum membrane, but at the same time may interact worse with the binding pocket of the COX protein itself. Taking into account that all the tested compounds have an effect on the lipid phase of membranes (in DSC studies), we decided to select those that interacted better with COX in the enzymatic assay for in-depth research. We have added a comment in the discussion section.

Very little comparison is done with the literature examples in the discussion. Please provide comparisons with current work on anti-inflammatory drug design, preferably multiple examples and with similar research methods such as phase transition analysis and COX inhibition studies. 

Zambrano et al. published a revision of using differential scanning calorimetry method in investigating drug-membrane interactions. They concluded that DSC method enables understanding the impact of studied compounds on lipid packing and stability of the phospholipid bilayer [doi:10.1016/j.bbrc.2024.149806]. Lichtenberger et al. conducted experiments using a model membrane made of DPPC and selected NSAIDs. The obtained results indicated that these drugs interact with non-charged phospholipids contained in the mucosa of stomach and intestines by forming complexes with them. This interaction may promote the gastrotoxic effect of those medicines [doi:10.1038/nm0295-154]. Lucio et al. showed that acemetacin, indomethacin and nimesulide influenced the phase transition temperature of model membrane. It was concluded that this effect may have a negative impact to the gastrointestinal mucosa, which is associated with the occurrence of side effects of these medicines [doi:10.1021/la703584s]. The thermal effects of selected oxicam derivatives were also evaluated by Kyrikou et al. using the DSC method. Their results were similar to ours. The perturbing effect of the drugs exerted on DPPC model membranes resulted in decrease of the main phase transition temperature and in broadening of the transition peaks [doi:10.1016/j.chemphyslip.2004.06.005]. We have added a discussion supported by literature. Changes have been highlighted in the text.

Same is true for introduction, very little current literature is discussed. Examples of novel anti-inflammatory compounds should be provided, and their pros and cons compared to the studied molecules should be discussed. This would significantly improve paper quality.

Where possible, newer literature has been introduced. The introduction aimed to present the problem of the urgent need for new anti-inflammatory drugs. For this purpose, coxibs were mentioned, which were promising as selective COX-2 inhibitors, but due to their side effects were removed from the market, which is why the 2005 publication refers to them. The discovery of the link between inflammation and cancer development is described next, to show that anti-inflammatory compounds are even more necessary. Then, it is indicated why 1,2-benzothiazine derivatives are considered promising new compounds with anti-inflammatory action and why certain chemical modifications were introduced. We believe that introducing a description of novel anti-inflammatory compounds would unfortunately take up too much space in the introduction, because this is a very broad topic, and the aim of the introduction is to signal the studies described in the subsequent publication. Whereas the comparison of new 1,2-benzothiazine derivatives with anti-inflammatory compounds was carried out in Chapter 4. Discussion.

What is a possible explanation of why compounds BS23 and BS30 more significantly affect lipid phase transition temperature, and demonstrate more beneficial docking energy (compared to other studied compounds), while the COX docking site is clearly in the middle of the protein, where direct interaction with lipid molecules is unlikely? Could there be correlation between the type of docking interactions demonstrated for each compound with its performance in inhibition of COX-2 or lowering of phase transition temperature? For example, Compound BS30, which is among the best in the study demostrated the least amount of van der Waals interactions and most amount of pi-type interactions. Is it related to the hydrophobicity of molecule moieties and type of incorporated functional groups? What could be said about poorly performing molecules, what are they lacking?

The ability to interact with phospholipids is related to the lipophilicity of the compound. The ability of the molecule to bind to the COX protein is due to its structure which allows it to be matched to the appropriate amino acids in the protein. They will result from other characteristics of the tested compounds. Both must be met for the molecule to be a good drug candidate. Some discussion has been added to the manuscript and highlighted in the text.

Few minor points:

Figure 6 is difficult to comprehend. What exactly is behind the term "relative transition temperature"? Is it a relative value? Why is the Y-axis scale in absolute degrees then? Why are error bars so tall? Error bars on figures 4,5 where, supposedly, parent data is shown, seem to be smaller. Also the Y-scale is chosen poorly, showing very little difference between bars.

It is a comparison of the main phase transition temperature for two phospholipids (DPPC and DMPC) with the addition of the tested compounds in a molar ratio of 0.1. The studies were conducted with one replication (2 samples for each lipid and compound), so the error bars may be higher than in the case of other studies (that were conducted in triplicate or more). This Figure shows whether a compound has a similar effect on both phospholipids or not. Typically, the impact with DMPCs that have shorter carbon chains (shorter hydrophilic tails) is stronger, and this is also evident in our study, as the reduction in Tm for DMPC with a given compound is greater than for DPPC.

Page 6 row 234, what is TMPD? 

Thank you for this valuable comment. This has been corrected.

N,N,N cent,N cent-tetramethyl-p-phenylenediamine (TMPD)

Some errors and typos: page 12 row 501 "fosfolipid"; page 21 row 779 and page 22 row 787: leftover strikethrough formatting.

We have made the corrections

Reviewer 3 Report

Comments and Suggestions for Authors

In the present MS, anti-inflammatory properties of eight 1,2-benzothiazine derivatives were synthesized and evaluated compared with meloxicam.

1. However, which one among these 8 derivatives was the best candidate compound?

2. In addition, as shown in Table 1:

“BS26 showed IC50 values of 128.73 ± 9.1 µM for COX-1 and 18.20 ± 4.3 µM for COX-2, with a selectivity ratio of 0.14, indicating greater selectivity for COX-2 than meloxicam. Additionally, this compound showed lower cytotoxicity compared to meloxicam, with an IC50 value of 203.28 ± 22.9 µM.

BS28 showed IC50 values of 95.88 ± 5.7 µM for COX-1 and 12.46 ± 1.9 µM for COX-2, with a selectivity ratio of 0.13, indicating a strong preference for COX-2 over COX-1 and better selectivity than meloxicam. Its MTT assay IC50 was 179.74 ± 13.6 µM.

BS29 presented IC50 values of 124.81 ± 6.7 µM for COX-1 and 17.80 ±2.1 µM for COX-2, with a selectivity ratio of 0.14, indicating higher selectivity for COX-2 compared to meloxicam. Its MTT assay IC50 was 232.51 ± 11.8 µM.”

Therefore, if BS29 has an acceptable selectivity ratio for COX-2?

Author Response

Reviewer 3#

We thank the reviewer for his critical comments. The present paper has been revised according to all Reviewer comments. Below is the list of changes introduced to the present version of the manuscript:

In the present MS, anti-inflammatory properties of eight 1,2-benzothiazine derivatives were synthesized and evaluated compared with meloxicam.

1.However, which one among these 8 derivatives was the best candidate compound?

Our goal was to obtain new compounds with anti-inflammatory activity. In this work, the test confirming anti-inflammatory activity is the cyclooxygenase inhibition test, especially isoform 2 (COX-2). Based on the inhibitory activity towards COX-2, the most active compound among the new 1,2-benzothiazine derivatives is compound BS23 with IC50 = 13.19 µM. The second important parameter of the action of anti-inflammatory compounds is the highest selectivity towards COX-2, i.e. the lowest inhibition of the COX-1 isoform, because this is the cause of the adverse effects of anti-inflammatory drugs. In terms of selectivity, compound BS23 is again the best, because it inhibits COX-1 with IC50 = 241.64 µM, i.e. the selectivity coefficient is very good (0.05). This means that at the doses used, this compound should inhibit COX-2 and not inhibit COX-1.

  1. In addition, as shown in Table 1:

“BS26 showed IC50 values of 128.73 ± 9.1 µM for COX-1 and 18.20 ± 4.3 µM for COX-2, with a selectivity ratio of 0.14, indicating greater selectivity for COX-2 than meloxicam. Additionally, this compound showed lower cytotoxicity compared to meloxicam, with an IC50 value of 203.28 ± 22.9 µM. BS28 showed IC50 values of 95.88 ± 5.7 µM for COX-1 and 12.46 ± 1.9 µM for COX-2, with a selectivity ratio of 0.13, indicating a strong preference for COX-2 over COX-1 and better selectivity than meloxicam. Its MTT assay IC50 was 179.74 ± 13.6 µM. BS29 presented IC50 values of 124.81 ± 6.7 µM for COX-1 and 17.80 ±2.1 µM for COX-2, with a selectivity ratio of 0.14, indicating higher selectivity for COX-2 compared to meloxicam. Its MTT assay IC50 was 232.51 ± 11.8 µM.” Therefore, if BS29 has an acceptable selectivity ratio for COX-2?

Yes, the BS29 compound not only has an acceptable selectivity ratio, but it is among the four best compounds in this range. These are the BS23, BS26, BS28 and BS29 compounds with a selectivity ratio of 0.05, 0.14, 0.13 and 0.14, respectively. We choose the best compound based on its activity against COX-2 and selectivity (COX-2/COX-1). Unfortunately, we had to choose only two compounds for further in-depth studies, therefore compounds BS23 and BS28 were chosen due to the best selectivity (0.05 and 0.13).

The MTT assay, on the other hand, tells us about the level of cytotoxicity of compounds against healthy cells. This test showed that most of the new 1,2-benzothiazine derivatives (except BS24 and BS30) are less toxic than the model meloxicam, because their IC50 is higher than the IC50 of meloxicam (174.23 µM). The most active compound against COX-2, BS23, also showed one of the lowest cytotoxicity (with IC50 = 228.40 µM) of the compounds in this group.

Round 2

Reviewer 2 Report

Comments and Suggestions for Authors

Although for all points the authors provided adequate and reasonable replies, including corresponding changes and additions in the paper manucsript, I cannot accept the reply to one question, regarding Figure 6. 

Below, find the copied question and reply from the first round of review:

Figure 6 is difficult to comprehend. What exactly is behind the term "relative transition temperature"? Is it a relative value? Why is the Y-axis scale in absolute degrees then? Why are error bars so tall? Error bars on figures 4,5 where, supposedly, parent data is shown, seem to be smaller. Also the Y-scale is chosen poorly, showing very little difference between bars.

It is a comparison of the main phase transition temperature for two phospholipids (DPPC and DMPC) with the addition of the tested compounds in a molar ratio of 0.1. The studies were conducted with one replication (2 samples for each lipid and compound), so the error bars may be higher than in the case of other studies (that were conducted in triplicate or more). This Figure shows whether a compound has a similar effect on both phospholipids or not. Typically, the impact with DMPCs that have shorter carbon chains (shorter hydrophilic tails) is stronger, and this is also evident in our study, as the reduction in Tm for DMPC with a given compound is greater than for DPPC.

It appears that the values given in this figure are relative compared to the Tm of pure lipid (black bar). Which means that sample Tm is related to Tm of pure lipid, so the degrees must cancel out. The Y-axis cannot be in degrees. Once again, the scale of this figure is not optimal (difference between bar heights is hard to see). No changes were done to the figure appearance or caption.

By reading the authors reply I can understand the main point of this figure and it does indeed show that generally the Tm shift is higher for shorter lipids (DMPC), but its not optimal, and it is confusing because of the Y-axis Please revise figure 6. I suggest using % instead of degrees and multiplying the axis by a factor 100. Also increase vertical scale, making it visible from 0 to 170 %, thus, the bars would be more differentiated, while all the bars and error bars would still be visible. Also provide notes what do the error bars correspond to: error, or standard deviation. Maybe S.d. could be used which is probably not that big.

Otherwise, the revised version is nice and the new discussion is interesting to read.

Author Response

It appears that the values given in this figure are relative compared to the Tm of pure lipid (black bar). Which means that sample Tm is related to Tm of pure lipid, so the degrees must cancel out. The Y-axis cannot be in degrees. Once again, the scale of this figure is not optimal (difference between bar heights is hard to see). No changes were done to the figure appearance or caption.

By reading the authors reply I can understand the main point of this figure and it does indeed show that generally the Tm shift is higher for shorter lipids (DMPC), but its not optimal, and it is confusing because of the Y-axis Please revise figure 6. I suggest using % instead of degrees and multiplying the axis by a factor 100. Also increase vertical scale, making it visible from 0 to 170 %, thus, the bars would be more differentiated, while all the bars and error bars would still be visible. Also provide notes what do the error bars correspond to: error, or standard deviation. Maybe S.d. could be used which is probably not that big.

Response

We thank reviver  for the tips on how to make our results clearer. We admit that the description of the axes in the graph was misleading, due to the fact that degrees Celsius divided by degrees Celsius cancel each other out, so the unit was dimensionless. We drew Figure 6 again according to the reviewer's instructions. We decided to remove error bars to improve image readability.